# Molecules, Information and the Origin of Life: What Is Next?

**DOI:** 10.3390/molecules26041003

**Published:** 2021-02-14

**Authors:** Salvatore Chirumbolo, Antonio Vella

**Affiliations:** 1Department of Neurosciences, Biomedicine and Movement Sciences, University of Verona, 37134 Verona, Italy; 2Verona-Unit of Immunology, Azienda Ospedaliera Universitaria Integrata, 37134 Verona, Italy; antonio.vella@univr.it

**Keywords:** origin of life, Shannon dissipation, dissipative system, molecules, Boltzmann, information, complexity

## Abstract

How life did originate and what is life, in its deepest foundation? The texture of life is known to be held by molecules and their chemical-physical laws, yet a thorough elucidation of the aforementioned questions still stands as a puzzling challenge for science. Focusing solely on molecules and their laws has indirectly consolidated, in the scientific knowledge, a mechanistic (reductionist) perspective of biology and medicine. This occurred throughout the long historical path of experimental science, affecting subsequently the onset of the many theses and speculations about the origin of life and its maintenance. Actually, defining what is life, asks for a novel epistemology, a ground on which living systems’ organization, whose origin is still questioned via chemistry, physics and even philosophy, may provide a new key to focus onto the complex nature of the human being. In this scenario, many issues, such as the role of information and water structure, have been long time neglected from the theoretical basis on the origin of life and marginalized as a kind of scenic backstage. On the contrary, applied science and technology went ahead on considering molecules as the sole leading components in the scenery. Water physics and information dynamics may have a role in living systems much more fundamental than ever expected. Can an organism be simply explained by a mechanistic view of its nature or we need “something else”? Probably, we can earn sound foundations about life by simply changing our prejudicial view about living systems simply as complex, highly ordered machines. In this manuscript we would like to reappraise many fundamental aspects of molecular and chemical biology and reading them through a new paradigm, which includes Prigogine’s dissipative structures and informational dissipation (Shannon dissipation). This would provide readers with insightful clues about how biology and chemistry may be thoroughly revised, referring to new models, such as informational dissipation. We trust they are enabled to address a straightforward contribution in elucidating what life is for science. This overview is not simply a philosophical speculation, but it would like to affect deeply our way to conceive and describe the foundations of organisms’ life, providing intriguing suggestions for readers in the field.

## 1. Introduction

The origin of life is a fundamental conundrum for science. The main texture of any theoretical tenet about life encompasses only molecules and energy, as enabled agents to start life on earth somehow, although the way by which this occurred is still under debate. How molecules and energy interacted to originate life? And moreover, did chaos, i.e., our common idea of “disorder” and complexity, i.e., our usual idea of “organized order”, the only forces building up the living world? [1,2,3,4,5,6,7]. All these questions are of the utmost importance in any debate about the “origin of life”. The role of molecules in bursting and supporting life is quite indisputable, inasmuch many important theories forwarded to address the topic, fund their tenets on the ability of molecules to build up a living system [1,2,3,4,5,6,7,8,9,10,11,12,13,14,15,16]. To highlight the way by which molecules started life, researchers are used to refer to abiotic principles, accounting on the simple role of chance and necessity [17] but often separating, in a Cartesian fashion, chemical masses from energy. The result of this intellectual habit leads people quite inevitably to read the natural world as a kind of “mechanism” [18]. Describing a living system as an organized, complex device compels researchers to suggest an “end orientation finality” in the debate about the origin of life, i.e., a sort of purpose or a final planning, something similar to what Jacques Monod called “teleonomy” [19]. The abiotic principle should have the ability to conceptually explain how to build up life from casuality and randomness, without a necessity-driven final task. According to some authors, for example, an open reaction network, even the simplest model made by non catalyzed secondary reactions, can spontaneously reach a stationary state, which then comes to possess self-organizing ability [20]. Self-organization is not driven necessarily by a teleonomy. Genial intuitions, therefore, moving intriguing experimental setting, are discussing about the possibility to explain self-assembling and self-renewal properties of a living system with few thermodynamic rules.

However, as life has the puzzling ability to self-replicating and self-organizing without an apparent ordered plan, a reappraisal on the role attributed to molecules and energy to highlight how life emerged should encompass a more dynamical, than mechanistic perspective, for example considering ordering, self-replicating mechanisms, such as dissipation, spontaneously emerging from the chaotic sink formed by the primordial soup. Actually, a major outstanding question we should address is “what is life” just in its core meaning. Life contains a huge component of chaos, therefore the idea of life as a mechanism might fail in its theoretical concept. Yet, a thorough definition is still far to be elucidated, if a mechanistic view holds the main tenet about its origin [14]. In the mechanistic view, for example, causes and effects are completely mandatory to describe the mechanism itself, whereas a dynamical perspective might describe the system by simply showing and evaluating changes in different states, aside from their beginning or purported end. In order to get a sound definition of life, we must completely revise our own concepts of what life really is, even affecting the orthodox biological tale about molecules in a living system described so far in the literature. Probably, the living system is all but an ordered, teleonomical-oriented device, it might be more properly a dynamics who continuously strives to be held as an “ordered” reality. A thorough definition of life is useful for philosophy but, in this context, it could give insights to many fields of biology, such as exobiology. Gerald Joyce from the Scripps Institute defined life as something able to self-organize and self-assembling and moreover whose evolution is handled by Darwinian natural selection [15]. The core conundrum of this definition is actually the self-organization. But why this occurs? 

A clear, sound definition of what fuels, supports and generates this self-assembling and self-organizing property, is yet far to be fully elucidated. Some researchers addressed this puzzling issue, without a true, encouraging success, yet. Maturana’s pioneering work on the concept of “autopoiesis” [21,22,23], which can be retrieved in very recent papers [22,23,24,25], introduced intriguing insights about how life is “auto-representing”. Maturana, however, stands on a plain philosophical ground. Researchers are still searching for an ideal theory based on biology about how life arose and why it appears in a completely distinct entity respect to crystals, despite its chemico-physical nature [26]. Alongside with the concept of “self-replication” and “self-organization”, characterizing the idea of what “a life is”, some controversial issue may occur, as a virus or a prion, for example, are not properly considered living systems, i.e., organisms, despite their self-replicating and self-organizing endowment. Therefore, life is more than a self-replicating object. A typical “structure” model was added to the ideal “functional” concept of life, e.g., the existence of separate compartments as hydrophobic membranes and the onset of a homeostatic mechanism [12]. This would suggest that the dynamics of life is much more complex than a simple chemistry-driven dissipation. Despite we can perfectly recognize a living organism from a mineral stone or a crystal, the most puzzling question regards how chemical molecules can distinctly form an inert crystalline object or alternatively a dynamic, self-assembling and self-organizing living system. Which is the “Maxwell’s evil” able to transform an inert chemical substance to a living object? Is the associated response still within raw chemistry and physics or we should seek for a kind of “spirituality” or “new physics” of the condensed matter, as Schrödinger provocatively suggested many years ago? [27]. 

We do not know. Most probably, biology has to restart from a new standing point, a novel epistemology, re-shaping our way to describe the innermost texture of what “is able” to generate something that can evolve as a “life”, i.e., a living system. We have to thoroughly revise our way of modeling the chemical and physical interpretation of natural phenomena, so to introduce new intriguing concepts able to shed light on a more appropriate concept of life. Which are the intellectual tools we can adopt to define life in its deepest meaning?

Actually, the simplest and commonest way to describe life, is a mechanistic perspective, borne from statistical chemistry, which funded many models able to elucidate the evolution of complex systems [28,29]. When thinking molecules in their abstracted form, usually we describe a series of perfectly equal particles freely interacting and moving each others because of the thermal convective forces and Brownian motion. The event space where this occurs can be named “Boltzmann space”. In the Boltzmann space, particles (molecules) have the highest degrees of freedom, representing what it can be called an “ideal” Boltzmann space, as in nature we can observe only a raw approximation of this circumstance. Molecules, in this abstract and ideal vision of the chemical-statistic world, are particles perfectly equal each others, sharing the same degrees of freedom, a condition we call “equisotropic Boltzmann space” (E-space). An E-space is very hard to really exist in nature but presumably it might occur if considering crystalline monomers, simple molecules or atoms in a Boltzmann space. Usually an E-space is an “ideal” events space, very close to a theoretical gaseous thermodynamic system. Conversely, a Boltzmann space made by particles each different from the others, is called a “disquisotropic Boltzmann space”(δίσ-“different”) or D-space. Briefly speaking, D-space and E-space are altogether named “isotropic Boltzmann spaces”. A D-space is made by particles endowed with complete uniqueness. This distinction is crucial to introduce and discuss the concept of chance and necessity in the origin of life we intend to address further on in this manuscript.

Chance and necessity are still considered fundamental engines of the autopoietic life, despite some controversial opinion about their antinomy in science [30]. The Monod’s chance-necessity model has shaped our ways to represent the concept of life no more no less than the idea of “nothing” in the concept of the Universe and its birth [31,32,33]. According to many renowned opinions, matter as well as energy arose from nothing, i.e., *ex nihilo* [33]. The concept of nothing we would like to push forward is only used as a foreword to explain further important concepts about our theses. The “nothing” may be even an ambiguous concept, anyway. Nothing might be trivially “a place” without “anything” (i.e., with none, which we call N0-space)) or, in a more tricky assumption, “something” having “no sense”, (a N*x-*space), i.e., nothing may be either a place completely empty or completely full and yet useless. This conceptual framework is fundamental to enable us, as biologists, in addressing new important speculations about life (see below).

## 2. Chance, Necessity, and Dissipation: The Concept of “Nothing”, “Nonsense”, Anisotropy, and End Orientation

### 2.1. Nothing (N0-Space), “Non Sense”(Nx-Space), E-Space and D-Space: Which Context Life Needs to Emerge and Evolve

Our introduction of concepts such as nothing, D-space and E-space, serves to weave a logical tale of how life most probably arose. 

Nothing is an empty space (N0-space). From a philosophical point of view, a N0-space cannot be the theoretical place from which life commenced. Therefore, when is claiming that life started from “nothing” [34], this nothing should be better identified. As a matter of fact, we are used to define “nothing” something lacking a sense, a meaning, a possible “space” full of objects without a “sense”, not simply devoid of a sense in the simplistic terms of a “meaning”. Sense can be interpreted as a synonymous of “orientation”, somehow intended as a vectorial addressing. Biology can have a “sense”, as an orientated target, an “end orientation”, because the chemico-physical phenomena underneath a biological system, as well as any further natural system, are moved by irreversible forces, with increasing entropy. Biologists, who would like to take into account the concept of “nothing” as the major source of matter and energy and hence of life, should be aware that “nothing” might not be simply the complete lack of “anything” (an N0-space) but, on the contrary, a crowded presence of elements without “a sense” (a N*x-*space) [34]. 

The concept of “nonsense” is usually misinterpreted in a teleonomical point of view, contrasting with an ordered idea of information [35]. Since the renowned sentence by Theodosius Dobzhansky “Nothing in biology makes sense except in the light of evolution” [36,37,38,39,40], the moving engine of a developing living object was the evolutive course of natural selection. This model considers that “outside”, Nature can shape chance, following necessity, to an increasing adjustment of complexity, i.e., the orchestrated functional order. The concept of sense affects deeply the role with which chance and necessity started life, because “sense” has been quite always misinterpreted with the concept of teleonomy, whereas it could mean simply “orientation”, something dealing with a dynamics, a movement [37]. Briefly speaking, in our model, the tern “sense” means a vectorial orientation (“end orientation”) without a final purpose (a teleonomy). The word orientation hides within the concept of “movement”, which is a much more intriguing and fundamental concept than commonly expected. Why we are talking about nothing, end orientation and movement? We trust these hallmarks are at the foundations of life. Actually, we can further elucidate this proscenium by an eye opening example.

A painting made by the simple, completely blank canvas can be considered a N0-space, i.e., with “none”, whereas a painting made by a highly crowded and raw jumble of color spots is an N*x-*space, as it has “no sense”, as it is meaningless either if the painting is within a correct orientation placement or alternatively is upside down. This circumstance can be also obtained if the painting is made by perfectly equal forms equally spaced each others (recalling an E-space). These examples are representations of three different start-points in the conceptual way to define how life started and what life is in its depth: 1) an event space without particles; b) a D-space; c) an E-space. The “nothing” on the canvas can be indifferently represented by a blank canvas or a completely uniform black canvas. On the contrary, if we spot ether the blank or the black canvas with a red stain, somewhere on the canvas, then the painting “earns” a movement feature, the painting is different depending on its orientation on the wall. Actually, we introduced a sort of “diversity” in the isotropic space, as well in the N*x*-space, which we call “anisotropy”and the whole painting gained a “moving” perception. This quite naïve example would like to introduce to the standpoint concept that at the “beginning” of life, anisotropy might have started an evolutive pathway. The explanation how anisotropy probably contributed in bursting life, can be the following. The appearance of a “diversty” in an E-space, should create anisotropic particles with different degrees of freedom respect to any E-particle, so creating a cascade of events with increasing probabilities to form complexes and hence an evolutive pathway. Probably, life started and evolved when a disquisotropic element (a “diversity”) occurred in an E-space. The model works only if “something” disturbs a “N*x-*space”, made by an E-space.

What we are trying to develop further with this speculative tale? 

An E-space is theoretically made by equal particles, which can undergo interactions leading to bonds and forming initial pairs. Particles with equal hallmarks and features (equisotropic) have the same probability to join as well as to detach, so they could not organize a developing system, as chance provides particles with the same probability to join or to detach each others. In this perspective, this E-space can be considered a “N*x-*space”, from an evolutive meaning. The introduction of a disquisotropic particle substantially modifies the “N*x-*space” to an evolutive dynamics; it introduces “anisotropy”, as the probability to form a pair E-D, is completely different from an E-E pair and can trigger further more complex interactions to be formed and to change dramatically the evolutive pathway of the whole system towards highest organization.

Starting from an ideal E-space, without replicating events, the evolutive route will reach inevitably a D-space, at last. The D-space is perfectly characterized by the existence, within the space, of elements completely or even slightly different each others, so to make us able in defining this space as the “space of the uniqueness”. At this point of our overview, no particle, based simply or solely on its disquisotropy is enabled to push ahead the evolutive system, because any particle within the system “might” do it, at least theoretically, and the system is blocked, from an evolutive orientation, it becomes a “nothing” (a N*x-*space), in evolutive terms.

The informative burden of an E-space is represented by one single highly reiterated information, and hence with the lowest Shannon’s entropy (it is similar to a single word correctly pronounced and amplified with a very loud voice), whereas the informational burden of a D-space has the highest Shannon’s entropy, being comparable to a crowded population of different and unique messages. The evolutive pathway of an E-space is hampered by probability, the one of a D-space by informativity. 

Yet, during the evolutive course, order can be formed. How does this happen? We will turn back to this concept further on, but order, with complexity, cannot avoid the final achievement of a “N*x-*space” formed by a D-space, if funded only on chance and necessity. This can be explained.

A possible calculation held on theoretical basis, showing that order changes the “chance-associated possibility”, can be attempted.
(1)$$C_{K}\left(n\right)\;=\;\frac{n}{k}\;=\;\frac{n!}{k!\left(n-k\right)!}$$
where *n* = 1000 e *k* = 2 and results in an approximate value of ≈499,500 combinations, without order and repetitions. The introduction of an order accounts on the following Equation (2):(2)$$V_{K}\left(n\right)=\;\frac{n!}{\left(n-k\right)!}$$

And results to ≈999.000–1,000,000 variations, including if containing repetitions Equation (3): (3)$$V_{k}^{\prime }\left(n\right)=\;n^{k}$$

An amount of 1000 particles is much too high to forecast a correct combinatorial value. If our particles moving and interacting in the Boltzmann space are only 21, then combinations *C* with *k* = 1, according to Equation (1) are 21, if *k* = 2, *C* = 210, if *k* = 3, *C* = 1330. With *k* = 3 introducing order, variations are 7980. Order increases the number of variability inside the combinatorial process, so increasing, as variability, the Shannon entropy. So, if order is formed by chance and necessity, variability, within a D-space, increases Shannon’s entropy and enhances the possibility to reach an N*x*-space, i.e., a non sense space coming from a complete D-space, if a dissipation mechanism does not occur (see below).

This speculative description of how a dynamical evolution could start only accounting on chance and necessity, gives us the opportunity to assess that even anisotropy, though providing an ignition to the engine of life, cannot sustain an evolutive path without further dynamics such as dissipation.

### 2.2. The Engine of the Dissipation. Shannon Dissipation

Dissipation is a dynamics that introduces replicative and reiterative mechanisms to build up inner order. Let’s come back, therefore, to the outstanding question: how life arose from the chemical crowded environment called prebiotic soup? 

Systems working within the thermodynamic balance and endowed of irreversibility can be simply considered as complex fluxes of matter and energy, where matter and energy enter the flux and degraded matter with heat leaves the flux, according to the second principle of thermodynamics. These are open, irreversible systems, working not far from the thermodynamic equilibrium and increasing entropy, whereas life dynamics works far from the thermodynamic equilibrium, though increasing as well total entropy outside the system [41,42,43,44]. Many attempts were made to explain why life self-organizes and self-replicates by acquiring increasing complexity and, at the same time, gaining the ability to “learn” from the environment. Aside from the conceptual framework held by Maturana and Varela [45], very recently Jeremy England proposed an evolutionary model based on a “dissipation-driven adaptation”, funding its theoretical thesis on a physical, spontaneous emergence of life, i.e., a statistical physics of the origin of life [46]. In his theory, molecules can self-assembly in a random way to better uptake free energy and dissipate heat, so spontaneously creating a complex and evolutive process [46]. The concept of self-assembling and self-organizing by using negligible amount of free energy, usually made available by mesoscopic arrangements in the surrounding water clusters, is particularly stressed in science [20]. England’s model is attractive but lacks of an evolutionary sense, as his model is restricted to the very close boundaries of the molecule and water nano-environment, without a “pushing on” dynamics. In our opinion, in order to follow an end orientation leading to an evolutionary process, “something” should dissipate not only heat but also the informational burden earned with the increasing tailoring of entropy in the macrostate. Interestingly, England dismisses the environment as an evolutionary shaper, accounting only on the simple dissipation-driven teleonomy, but, in conclusion, despite its fascinating perspective, the simple statistical dynamics may in any time be halted in a “non-sense” auto-dissipating thermodynamic system. A possible solution to the puzzling issue of teleonomy in the living being is to greatly reduce the contribution of chance in the evolutionary process, by simply considering some “constant” dynamics. Actually, this may be met only with a dissipative mechanism.

Dissipation is a concept born with the Nobel Prize for Chemistry Ilya Prigogine, which is widely used in many ecodynamic models, including life thermodynamics [47,48,49,50,51,52,53]. Fundamentally, a dissipative system is a thermodynamic irreversible system operating out from the thermodynamic equilibrium, which is endowed with the ability to drive itself towards a critical state, a tendency called self-organized criticality (SOC) SOC is a hallmark that will originate avalanches of actions starting from fluctuations [54], although SOC is a state occurring in thermodynamic systems, such as living objects, only in certain conditions [55]. Chemical examples of dissipative systems are intriguing models such as the Bénard-Rayleigh cells and the Belousov-Zhabotinsky reaction, which are fundamentally characterized by cyclic reiterations of a kind of pattern [56]. Moreover, reiterations or replications, are roughly constant dynamics and also fundamental features of the autopoietic mechanism of life, therefore, some researchers suggested the hypothesis that life started fundamentally as a dissipative system, at least at its microstate [57,58]. Interestingly, at this level, Boltzmann and Shannon share comparable laws in the entropic phenomena, as the mathematical interpretation of thermodynamic entropy established in the statistical physical chemistry by Ludwig Boltzmann and J. Willard Gibbs in 1870 is similar to the information entropy developed in the forties of the XX century by Claude Shannon and Ralph Hartley, i.e.,
(4)$$S=-k_{B}\underset{i}{\sum }p_{i}\ln p_{i}$$
where *S* is the Boltzmann’s entropy and *p_i_* is the probability of the microstate at the equilibrium, is similar to
(5)$$H\;=-\underset{i}{\sum }p_{i}\log_{b}p_{i}$$
where *H* is the Shannon’s entropy and *p_i_* is the probability of a defined message in the Shannon space [59]. However, this does not mean that at the macrostate level, in an active dissipative system, entropies of Boltzmann and Shannon cannot behave in a quite opposite ways. The close relationship between thermodynamics, which is underneath the Prigogine’s dissipative mechanics and information, probably represents the core meaning of how life evolved in the physical nature and henceforth this Boltzmann/Shannon orchestrated relationship should be our pathway on which attempting a novel model to “explain” life. In this context, we would like to recall, for example, the Szilard’s model of the Maxwell’s demon scenario where it appeared clear that changes in information have thermodynamic consequences [60,61]. If life emerges as a dissipative phenomenon [48], the relationship between Boltzmann and Shannon should involve macrostates, more than microstates. With the aim to elucidate how a chemical dissipative structure evolved towards an end oriented course, giving rise to an evolutive system, it is speculatively possible that information exerted a major, outstanding role in the whole developmental process. One of the most striking macroscopic aspects of a dissipative system is its pattern-reiteration, as exemplified by Turing patterns in morphology, e.g., Bènard cells (such as biological cells) or Belousov-Zhabotinsky oscillations (such as many chaotic signaling pathways in a biological system) [62,63]. Organisms are fully characterized by replicated mechanisms and replication is a deep characteristic of living systems. In our thesis, replication may have a fundamental purpose also for Shannon entropy, besides to occur in a Boltzmann space because of a dissipative mechanism. 

If replication occurs in an anisotropic space, then Boltzmann increases its entropy (more particles added to the system, each with its degrees of freedom), whereas Shannon decreases its entropy, as a “single” message is highly reiterated, i.e., amplified. A simple explanation of this phenomenon can be gained by having a look to the Shannon’s entropy equation:(6)$$H\left(X\right)=\;-\overset{N-1}{\underset{i=o}{\sum }}p_{i}\log_{2}p_{i}$$

Taking into account a string of 21 different symbols, such as for example alphabetic letters in a European (Italian) speaking language, Shannon’s entropy is particularly high (*H(X)* = 4.39). Different symbols may exemplify a D-space. If, in the same space, we introduce a replication of a single, specific symbol, i.e., the letter “a” five times within a total number of 21 symbols (letters), then *H(X)* is reduced to 3.74. Again, with 10 replicated “a” on a total of 21 letters, *H(X)* = 2.95, with fifteen replicated “a”s *H(X)* = 1.60, with 20, *H(X)* = 0.28. Replication of a part in the Boltzmann space (amplification) reduces Shannon entropy in its space. 

At this point of our discussion, question arose why biological organisms are spontaneously induced to replicate parts of their dissipative and chaotic systems and why an up-growing complexity is formed. We can attempt a possible explanation.

Anyone can observe the undisputable reality that replicated things in the biological nature are not “perfectly” similar. Therefore, the apparent reduction in the Shannon’s entropy due to replication of the same information, collected in a defined structure, for example a leaf, holds a certain degree of intrinsic increasing entropy, in the sense that more replications are made in the event space, more “imperfections” are added to the incoming E-space, endowing equisotropic particles “with” a final intrinsic “slight” disquistropy. Briefly speaking, despite the fact that leaves from a plane tree are typical of a plane tree and absolutely different from an alder or a fir, each leaf from a defined tree is quite dissimilar within the same tree and from any other one within the same species, though minimally. Furthermore, replications in the Boltzmann space, which occurs via a dissipation mechanism, releases entropy in the environment, as it must fundamentally obey to the Second Law of Thermodynamics, despite creating order inside. The release of increasing entropy in the environment is highly facilitated, as the external world is much wider and bigger than a biological object. The increase in entropy outside the system pushes onward the system to go ahead in replicating patterns and creating order [64,65,66]. In this sense it goes on dissipating contrasting forces until the system is “frozen” (somehow) in a highly ordered state, where, more frequently, it starts to operate not far from the thermodynamic equilibrium. If this occurs, then order may even undergo a degradation event, as subjected to the thermodynamic second law. In both cases, the system is blocked: either to an excessive order or to a disordered chaos, without an evolutive pathway. In this circumstance, the object might be very similar to a “not living” thing, as it has stopped its dissipative engine. Therefore, something “inside” the dissipative system, a system held by thermodynamics, should keep working the dissipative machines aside from the increasing complex order.

We are emphasizing the concept that self-replication by dissipation, in biology, is absolutely fundamental to distinguish a living thing from any other ordered object present in nature, which can be highly ordered (crystal) or disordered (amorphous state). Replications have a defined informational endowment in a dissipative system; they are not simply ordered things coming from thermodynamic (Boltzmann) dissipation but are major hallmarks of the Shannon space. Yet, whereas in the Boltzmann space the outside increase in entropy pushes onward the creation of an inner order, the “hidden” increase in entropy in the Shannon space, we called dissipative disquisotropy, occurs “inside” the system and therefore forces the system to be continuously dissipated from the inside. Briefly speaking, while during the replication of a component in the Boltzmann space, Shannon entropy is decreasing, at the same time an underneath increase in disquisotropic entropy creates a sink of positive Shannon entropy that compels the system to restart in its dissipation dynamics. This kind of dissipation allows the system in “staying alive”, whereas Boltzmann dissipation in “going ahead” towards an evolutive complexity.

Figure 1 tries to summarize these points.

Let’s try to explain Figure 1.

Most probably, we are initially in an anisotropic event space where both Boltzmann and Shannon spaces have high entropy. Replication of a defined particle, cycle or pattern occurs in this circumstance (Figure 1, point 1A, amplified replication), leading to a decrease in Shannon’s entropy “but” an increase in Boltzmann’s entropy, which thereby is compelled to reduce this entropy, for example by creating polymers (Figure 1, point 1B). Polymers, as well as dissipative disquisotropy increase Shannon’s entropy (Figure 1, point 2A), due to their combinatorial variability, a circumstance that compels Shannon to replicate polymers (Figure 1, point 2B) and then obliging again Boltzmann to join polymers or macrostructures forming complexity in order to reduce its entropy (Figure 1, point 3A). At this point, both Boltzmann and Shannon have reduced entropies but Boltzmann must obey to the second principle of thermodynamics by releasing degraded matter (Figure 1, point 3B). This should restart the whole orchestrated Boltzmann-Shannon dissipative system. If the need of dissipating contrasting forces in the thermodynamic flux, can be called “end orientation”, as a recognized feature of the Boltzmann space, the same should be considered for the Shannon space. In few words, biological systems stay alive and go ahead in their “autopoietic” intrinsic hallmarks thanks to the Shannon dissipation. This speculative thesis we are forwarding so far, may have a rational ground in Szilard’s thesis and Toyabe et al’s evidence [60,62], addressing the ability of information (Shannon) to shape thermodynamics (Boltzmann). It is presumed that organized water inside the biological texture, via its mesoscopic rearrangement, even with the contribution of hydrophobic pockets, have linked Boltzmann dissipation to changes in information, leading to a kind of dissipation in the informational burden of the living object. Our hypotheses is that the orchestrating mechanisms we described above (see Figure 1) may be organized and moved on by biological, intracellular and confined water [67,68,69,70,71].

## 3. Starting with Complexity: From Early Life to Complex Living Things

### 3.1. How Complexity Probably Arose. The Concepts of Degree of Freedom, Relation and Memory

So far, we have highlighted the fundamental role exerted by two intertwined dissipative systems the Boltzmann’s and the Shannon’s one, in moving onward, along the end orientation, the dynamics of organisms. Complexity has been defined as such product finally rising up from entropy reduction of increasing order. Fundamentally, in our overview of how life evolved, complexity, likewise order, may not necessarily have an intrinsic teleonomy. It arises from the simple need to dissipate felt by the living system [72]. The most widespread idea of complexity may include the concept of regularity [73], therefore a complexity should be necessarily an ordered product coming from Shannon dissipation, as replication is associated with Shannon entropy reduction. In biology, new models have been proposed to address the concept of complexity, as the major concern was to overcome the concept of reductionism [74,75,76]. Fundamentally, the idea to read and understand complexity by an “omic” perspective, should include undoubtedly a teleonomical vision, as without a final functional purpose the same interpretation of the whole dynamic network fails abruptedly without a meaning. Our consideration is fundamentally opposite: complexity arises as a dissipative byproduct, likewise order, its teleonomy is probably the consequence of puzzling mechanisms we are going to discuss in this paragraph [77].

Taking into account the synopsis held by Figure 1, the “threshold” beyond which a dissipative system stops to be a chemical-funded mechanism and become a biological system, may be the increased rate of replication Figure 1, point 1A) respect to degradation Figure 1 point 3B), a circumstance that could be aided, for example, by the ability of lipids and hydrophobic pockets in molecules to change water structure. 

Turning again to the event space where our fascinating history took place, a first distinction has to be made, i.e., between interactions and relations. Two particles in the Boltzmann space are in relation (related) when their mutual interactions change their respective degrees of freedom. This startpoint is important to introduce a possible explanation of why complexity arose in the evolutive history. Fundamentally, two interacting particles have maximal degrees of freedom, respectively, as freely moving and interacting in the Boltzmann space. When their interactions change the degree of freedom of each participating particle, then we can reach a “relation”. However, no particle is theoretically willing to lose its degree of freedom; therefore a relation can be formed “only” when the degree of freedom of a pair is higher that the single particle alone.This “compulsion” can be triggered by Shannon dissipation, as the “gain” in degree of freedom is quite only of informative nature. We can argue, therefore, that complexity, with an informative teleonomy, arises once Shannon dissipation occurs, never before. As the gain in degree of freedom “inside” the ordered complexity is the engine with which complexity grows up, a continuous informational shaping is held throughout any further Shannon dissipation, so to lead to a “kind” of complexity we categorize as “teleonomical” or “finally end oriented”. The relation emerges with a dissipative dynamics, when replications and Shannon dissipation, put those relations to be joined in a complex structure and/or function (see again Figure 1). Yet, the relation is also the foundation of the concept of information. An object is not informative *per se* but if “related” to “other”. A glass is used to drink but it can be used to collect sand, to make pleasant sounds, to help drawing a circle on a paper and so on. The information “glass” cannot be merely defined by its chemistry or shape or use: it “depends”, despite we know very well what a glass is. Therefore, if is informative only an event which is “related” to a dynamics, the “memory” is its reiterated form, therefore. Memory is not a collected file or a constant and fixed object but reiterated (replicated, amplified) information.

### 3.2. Some Theoretical and Speculative Consequences

So, early life arose once a Shannon dissipative mechanism occurred. However, we are unable, so far, to ascertain when a dissipative, chemical system turns into a biological evolutive system, i.e., when Shannon dissipation is set up in the dissipative system, so starting an end oriented dynamics called life. We have forwarded some speculative ideas, on the basis of existing literature in the field, but further insights are needed. Is there a “threshold” for which a dissipative system made by inorganic matter evolves towards a living object? We forwarded some hypotheses but they remain speculative. Most probably, this might occur when the level of disquistropy inside a Boltzmann space overcomes the ability of the system to change this latter into an equisotropic space, or, more simply, when more complex molecules appear in the prebiotic soup and much more dissipative disquisotropy can happen [78,79]. We do not know, yet this might occur only if replication overcomes degradation. Taking into account that prebiotic soups were observed in further solar bodies besides earth [80], asking what life is still represents a straightforward and crucial topic for modern science. 

In conclusion, according to our speculative theses, life emerged as a chemical-physical dissipative system, where mineral, clays, water started a first catalytic reactions, then a dissipative mechanism and later a Boltzmann/Shannon orchestrated dissipation. Only when a kind of dissipation we called Shannon (or informational) dissipation was set, the chemical dissipative system turned onto a biological evolutive system, so generating order and complexity [77]. This complexity is not endless, anyway. Complexity, as outlined in Figure 1, should be “replicated” by Shannon. Therefore, the ability of a biological system to “limit” to a certain degree of complexity depends on the ability of Boltzmann to sustain the dissipation as a whole, taking into account the second principle of thermodynamics. This might suggest how come earth living organisms includes a widest range of complex living things from bacteria to humans. A kind of dynamic balance between Boltzmann and Shannon may explain why in the living nature do exist together and indifferently bacteria and whales. 

## 4. Limitations of the Study

This “perspective” article was attempted to provide novel suggestions on a currently debated topic, which is still far to be fully elucidated, anyway. We focused onto fundamental keystones in the discussion about the origin of life, being absolutely aware that much more insights are needed, particularly by modeling sound mathematical approaches and experimental data able to highlight our speculations further. The question of this kind of attempt would remain on that of cause and effect, but seeking for a “cause” and an “effect” might be particularly hard in this context. Unless the precise cause and its effects are unambiguously identified, the topic remains speculative, but probably necessary, at this time, to move an original suggestion: it may burst a fundamental discussion about autopoiesis. The question on the origin of life should be reduced to the question: what exactly is the cause and what are their effects, in other words, the origin of causes and effects. This might appear as a crucial weakness in this paper. However, the causes of Shannon dissipation were somehow indicated but elucidating this mechanism needs much further insights to be earned. Modern physics seems to work deeper and deeper in the quantum mechanical formulation of reality, also the macroscopic reality, where the identification of clear causes and clear effects is ultimately impossible, if not at least fundamentally blurred.

## 5. Conclusions

What is life? The gigantic question put by Schrödinger several years ago seems to lack a thorough and exhaustive definition, at last. In recent years, many fundamental theories have been reported and probably life emerged not merely by the random ability of chance and the mandatory pulse of necessity but by dissipative mechanisms, where probably information and thermodynamics are closely intertwined in a chaotic dynamic ultimately held by water. In this focused review, we tried to shed light on new incoming theories about the origin of life, hoping they find an attractive interest by readers engaged in this fascinating field.

## Figures and Tables

**Figure 1 molecules-26-01003-f001:**
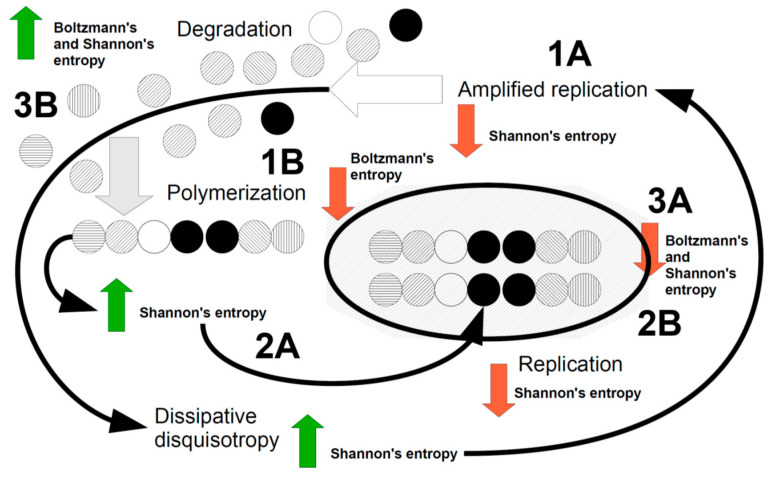
Cartoon showing the major dynamical events representing a living thing. The dissipative mechanism, in order to dissipate contrasting forces, replicates structures and/or functions (1A), usually increasing the E component of the Boltzmann space and reducing (red arrow) Shannon entropy. The energy is fueled by dissipation itself, via a rearrangement in the mesoscopic structure of water and of confined water. Replication may regard a single component in the Boltzmann space for example a single molecule or a single cell (amplification). Replication increases Boltzmann entropy, therefore the thermodynamic dissipation joins together elements in the Boltzmann space (energy fueled comes from dissipative mechanisms), creating polymers, which in turn increases Shannon entropy (green arrow) (2A). Polymers are joined to form macrostructures and more complex objects, so starting complexity, where both Boltzmann and Shannon entropy decrease (3A). At this stage, if degradation does not occur, (3B) the system might be halted and blocked. The increase in both Boltzmann and Shannon entropy restart the cycle. The energy for this step is given by thermodynamics (for Boltzmann) and the increase in disquisotropy associated with replication (1A), as any replicating event contains “imperfections”. This “dissipative disquisotropy” pushes Shannon to reduce entropy by replicating again and fueling the dissipation dynamics of the whole living system.

## Data Availability

Not applicable.

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
