# Peer review of "Molecules, Information and the Origin of Life: What Is Next?"

_molecules, 2021, doi:10.3390/molecules26041003_

Round 1

Reviewer 1 Report

This is an interesting, speculative attempt to explain life using concepts of dissipation, order, complexity and information, including an attempt to connect the Boltzmann entropy with the Shannon entropy. However, I found it to be rambling and not focused, and I was not able to logically follow the arguments made. The presentation really has to be improved before I can even judge the soundness of their arguments. 

A few other specific comments: I would add some references, including the NASA attempts to define life (see, e.g., Benner, Astrobiology 10 (2010), as well as other articles in this special collection edited by Deamer); published references for Jeremy England; and an interesting recent paper by Mayer on Order and Complexity in Life 10, 10 (2020).

I would use the term “Boltzmann space”, not a “Boltzmann’s space".

The authors introduce several new terms that are confusing and probably unnecessary, like sintropy and disquisotropy.

Additionally, the authors discuss known concepts without defining or explaining them, like teleonomy and complexity.

Author Response

Reviewer 1

This is an interesting, speculative attempt to explain life using concepts of dissipation, order, complexity and information, including an attempt to connect the Boltzmann entropy with the Shannon entropy. However, I found it to be rambling and not focused, and I was not able to logically follow the arguments made. The presentation really has to be improved before I can even judge the soundness of their arguments. 

Authors’ reply: Many thanks for this comment. The presentation was revised accordingly

A few other specific comments: I would add some references, including the NASA attempts to define life (see, e.g., Benner, Astrobiology 10 (2010), as well as other articles in this special collection edited by Deamer); published references for Jeremy England; and an interesting recent paper by Mayer on Order and Complexity in Life 10, 10 (2020).

Authors’ reply: Many further references were added accordingly

I would use the term “Boltzmann space”, not a “Boltzmann’s space".

Authors’ reply: Done

The authors introduce several new terms that are confusing and probably unnecessary, like sintropy and disquisotropy.

Authors’ reply: These terms are necessary for our purposes, yet we added explanations for their correct use

Additionally, the authors discuss known concepts without defining or explaining them, like teleonomy and complexity.

Authors’ reply: Those terminologies were further explained

Reviewer 2 Report

My research field is not physics but biochemistry or molecular biology. However, I am very interested in the studies on the origin of life from a standpoint of physics too, as described in the manuscript, because I want to consider the origin of life from the wide range of standpoints. Then, I would like to review the contents of the manuscript from such viewpoints.

Major comments

  1. First of all, it was difficult to understand Figure 1, because explanation in the legend of Figure 1 is quite insufficient for me as described below.

  (1) For example, did replication occur without supply of energy at 1A?

  (2) Should the dissipative disquisotropy in Figure 1 be an event at 2A?

 (3) What do the two arrows inside mean (around Shannon’s entropy: 2A)?

  (4) Should enough energy be supplied, which can compensate the increase of Boltzmann’s entropy at 3B?

Otherwise, degradation should occur more preferentially than replication

  1. Decrease of Shannon’s dissipation entropy should not always induce generation of information, because DNA/RNA with an ordered sequence does not always express genetic information for protein synthesis.

For example, genetic information is certainly composed of an ordered nucleotide sequence to express amino acid sequence information. However, DNA/RNA with an ordered sequence cannot always express a genetic information, because genetic information must encode amino acid sequence of a protein under the genetic code. In addition, genetic information should be formed accompanied by selection of a protein with a catalytic activity higher than others.

   Nevertheless, the three factors, protein, genetic information and selection, which are indispensable to create genetic information and to solve the origin of life, are not included in the model drawn in Figure 1. Therefore, the authors should discuss the validity of the model in more detail, which is drawn in Figure 1, that is, the reason why the origin of life can be reasonably explained according to the model drawn in the absence of the three factors.

Minor comments

  1. Legend of Figure 1 should be described below the Figure.
  2. Indention in the text is inserted incorrectly, such as lines: 136, 152, 161, 176, 237, 266, 278, 302, 316, 338, 348, 362.
  3. Unncessary indention is found at some places, such as lines: 164, 165, 167, 169-178, 179.
  4. Line 336: 3.1 should be 3.2.
  5. Italic characters (author’s names) on line 457 and 473 should be written in simple characters.

Author Response

Reviewer 2

My research field is not physics but biochemistry or molecular biology. However, I am very interested in the studies on the origin of life from a standpoint of physics too, as described in the manuscript, because I want to consider the origin of life from the wide range of standpoints. Then, I would like to review the contents of the manuscript from such viewpoints.

Major comments

  1. First of all, it was difficult to understand Figure 1, because explanation in the legend of Figure 1 is quite insufficient for me as described below.
  • For example, did replication occur without supply of energy at 1A?

Authors’ reply: At least theoretically, replication is a mechanism of the dissipative dynamics.  therefore energy for replication comes from dissipation. This supply is cumulated in mesoscopic changes of water free energy, giving molecular structures the possibility to work accordingly.  A sentence was added in the Figure legend.

  • Should the dissipative disquisotropy in Figure 1 be an event at 2A?

Authors’ reply: In 2A, the reduction of entropy in the Boltzmann space, creates polymers, i.e. linked chains of variable monomers, therefore polymers have a high Shannon entropy. Joining polymers to form macrostructures, macromolecules or organelles, reduces Shannon entropy. The dissipative disquisotropy occurs in 1A, as replications are not exactly and perfectly equal but contain “imperfections”, i.e. a disquisotropy. Figure legend was revised accordingly.

  • What do the two arrows inside mean (around Shannon’s entropy: 2A)?

Authors’ reply: upward green arrows mean increase. Figure legend was revised accordingly

  (4) Should enough energy be supplied, which can compensate the increase of Boltzmann’s entropy at 3B?

Otherwise, degradation should occur more preferentially than replication

Authors’ reply: This is a very good question. If the system is not maintained far from the thermodynamic equilibrium, degradation occurs more preferentially that polymerization and replication, i.e. the formation of order stops as the system is close to the thermodynamic equilibrium and behaves as an inanimate object. The dissipation dynamics is continuously fueled by the fact that the system is an open system far from the thermodynamic equilibrium and moreover because Shannon stores a huge sink of entropy within the disquisotropy, compelling the system to reduce Shannon entropy by replicating, amplyfing single molecules or particles, joining functionally structures to reduce their informational burden.

This part was added in the Figure legend

  1. Decrease of Shannon’s dissipation entropy should not always induce generation of information, because DNA/RNA with an ordered sequence does not always express genetic information for protein synthesis.

Authors’ reply: The generation of DNA and RNA polymers is a product of Boltzmann entropy dissipation

For example, genetic information is certainly composed of an ordered nucleotide sequence to express amino acid sequence information. However, DNA/RNA with an ordered sequence cannot always express a genetic information, because genetic information must encode amino acid sequence of a protein under the genetic code. In addition, genetic information should be formed accompanied by selection of a protein with a catalytic activity higher than others.

Authors’ reply: When we are talking about Shannon information we ae not dealing with the informational code of nucleic acids and proteins, we are talking about information as a whole, in a more general way.

   Nevertheless, the three factors, protein, genetic information and selection, which are indispensable to create genetic information and to solve the origin of life, are not included in the model drawn in Figure 1. Therefore, the authors should discuss the validity of the model in more detail, which is drawn in Figure 1, that is, the reason why the origin of life can be reasonably explained according to the model drawn in the absence of the three factors.

Authors reply: This Reviewer is right, we must detail better Figure 1 to prevent equivocal issues.

Minor comments

  1. Legend of Figure 1 should be described below the Figure.

Authors’ reply: Done

  1. Indention in the text is inserted incorrectly, such as lines: 136, 152, 161, 176, 237, 266, 278, 302, 316, 338, 348, 362.

Authors’ reply: indentions completely removed

  1. Unncessary indention is found at some places, such as lines: 164, 165, 167, 169-178, 179.

Authors’ reply: Removed

  1. Line 336: 3.1 should be 3.2.

Authors’ reply: Done

  1. Italic characters (author’s names) on line 457 and 473 should be written in simple characters.

Authors’ reply: Any Italic character removed

Reviewer 3 Report

This is a well written "perspectives" article on a currently debated topic. It operates mainly with the often cited analogy between the formula of the Botzmann entropy and that of Shannon's information transfer (Shannon entropy). It boils down to the organization of water molecules in living systems/organisms. The question of this kind of approach remains on that of cause and effect. Unless the precise cause and its effects are unambiguously identified, the topic remains speculative, but necessary. The question on the origin of life can be reduced to the question: what exactly is the cause and what are their effects, in other words, the origin of causes and effects. The authors propel the cause being the dissipation of Shannon entropy, which they show in an authoritative way, yet without proving their cause, or actually without distinguishing clearly causes and effects, which is ok, since they write that their standpoint is speculative. Modern physics seems to work deeper and deeper in the quantum mechanical formulation of reality, also the macroscopic reality, where the identification of clear causes and clear effects is ultimately impossible, if not at least fundamentally blurred.

Missing:

It seems approriate to cite, in addition to Rizotto [ref 7], a very useful and quite recent book chapter on the compilation and discussion of all definitions of life found in the literature: Kolb, V. Defining Life. In: Handbook of Astrobiology; Kolb, V., Ed.; CRC Press, Taylor & Francis: Boca Raton, FL, USA,
2019; Chapter 2.1; pp. 57–64. https://www.taylorfrancis.com/books/handbook-astrobiology-vera-kolb/e/10.1201/b22230

In the final part of section 1 (lines 91-95) the fact that nothingness can mean two fundamentally different things is regrettable, in the sense that the literature deserves these two terms to be clearly distinguished already in its name/term, rather than having to explain each time, as in the first paragraph of section 2 (lines 101-107).

The "naïve example" on what was "presumably" the "beginning of everything" is a bit too light, it should be elaborated deeper than just stating: "Probably, this example may work better if a disquitropic element occurs in E-space" (liney 129-134).

Already in the citations [refs. 2-9], but even more so during the discussion of choice of independent components, there is a recent work missing that does not exactly focus on replication, rather more on the importance of translation, however also from an information dissipating point of view, that should be cited: P. Strazewski. Low-digit and high-digit polymers in the origin of life. Life 9:17 (2019); https://doi.org/10.3390/life9010017. In that work, the multiplicity of digits correspond to the k value (number of distinct distinguishable entities) here.

The first time when the fundamental question on cause and effect arises comes with the statement (line 191): "In this theory, molecules can self-assemble (rather than self-assembly) in a random way TO BETTER UPTAKE AND DISSIPATE HEAT [my emphasis], so spontaneously creating a complex and evolutive process [34]." This presumes directly without words that the heat dissipation is cause rather than effect. Why? How can this be proven?

The very useful "simulated" calculation on the decreasing value of Shannon entropy with continuous replication should be preceeded/compared with an analogous calculation exemplifying step-by-step (as for the Shannon entropy) the increasing value of Boltzmann entropy with continuous replication, rather than just noting: "more particles added to the system, each with its degree of freedom".

At the end of section 2, where intracellular confined water is mentioned [refs 55-57], the works by Spitzer should be cited: J. Spitzer. From water and ions to crowded biomacromolecules: In vivo structuring of a prokaryotic cell. Microbiol. Mol. Biol. Rev. 75: 491–506 (2011), and J. Spitzer & B. Poolman. The role of biomacromolecular crowding, ionic strength, and physicochemical gradients in the complexities of life’s emergence. Microbiol. Mol. Biol. Rev. 73: 371–388 (2009).

The first sentence in section 3 (lines 302-304) reads as a somewhat too simplistic view: "... the dynamics of a living object, called organism." Since the living organism is intricately connected to its environment, which has been discussed previously, one would expect here the mentioning of at least the interaction of this organism with its environment, if not replacing the living organism with a living ecosphere.

Minor:

Abstract, line 14: theses (not thesis)

Line 67: We have to thoroughly revise (rather than has to)

Line 105: matter (rather than matters)

Line 169: is much too high (rather than is too much high)

Line 293: It is presumed that (rather than presumable)

Author Response

Reviewer 3

This is a well written "perspectives" article on a currently debated topic. It operates mainly with the often cited analogy between the formula of the Botzmann entropy and that of Shannon's information transfer (Shannon entropy). It boils down to the organization of water molecules in living systems/organisms. The question of this kind of approach remains on that of cause and effect. Unless the precise cause and its effects are unambiguously identified, the topic remains speculative, but necessary. The question on the origin of life can be reduced to the question: what exactly is the cause and what are their effects, in other words, the origin of causes and effects. The authors propel the cause being the dissipation of Shannon entropy, which they show in an authoritative way, yet without proving their cause, or actually without distinguishing clearly causes and effects, which is ok, since they write that their standpoint is speculative. Modern physics seems to work deeper and deeper in the quantum mechanical formulation of reality, also the macroscopic reality, where the identification of clear causes and clear effects is ultimately impossible, if not at least fundamentally blurred.

Authors’ reply: This comment is particularly appropriate. Therefore, we added some parts of it in the Limitations of the study

Missing:

It seems approriate to cite, in addition to Rizotto [ref 7], a very useful and quite recent book chapter on the compilation and discussion of all definitions of life found in the literature: Kolb, V. Defining Life. In: Handbook of Astrobiology; Kolb, V., Ed.; CRC Press, Taylor & Francis: Boca Raton, FL, USA,
2019; Chapter 2.1; pp. 57–64. https://www.taylorfrancis.com/books/handbook-astrobiology-vera-kolb/e/10.1201/b22230

Authors’ reply: Done

In the final part of section 1 (lines 91-95) the fact that nothingness can mean two fundamentally different things is regrettable, in the sense that the literature deserves these two terms to be clearly distinguished already in its name/term, rather than having to explain each time, as in the first paragraph of section 2 (lines 101-107).

Authors’ reply: Ok, the term nothingness was removed and replaced with other terminologies

The "naïve example" on what was "presumably" the "beginning of everything" is a bit too light, it should be elaborated deeper than just stating: "Probably, this example may work better if a disquitropic element occurs in E-space" (liney 129-134).

Authors’ reply: This part was revised, accordingly

Already in the citations [refs. 2-9], but even more so during the discussion of choice of independent components, there is a recent work missing that does not exactly focus on replication, rather more on the importance of translation, however also from an information dissipating point of view, that should be cited: P. Strazewski. Low-digit and high-digit polymers in the origin of life. Life 9:17 (2019); https://doi.org/10.3390/life9010017. In that work, the multiplicity of digits correspond to the k value (number of distinct distinguishable entities) here.

Author’s reply: This reference was cited (ref n. 35.)

The first time when the fundamental question on cause and effect arises comes with the statement (line 191): "In this theory, molecules can self-assemble (rather than self-assembly) in a random way TO BETTER UPTAKE AND DISSIPATE HEAT [my emphasis], so spontaneously creating a complex and evolutive process [34]." This presumes directly without words that the heat dissipation is cause rather than effect. Why? How can this be proven?

Authors’ reply: This part was revised

The very useful "simulated" calculation on the decreasing value of Shannon entropy with continuous replication should be preceeded/compared with an analogous calculation exemplifying step-by-step (as for the Shannon entropy) the increasing value of Boltzmann entropy with continuous replication, rather than just noting: "more particles added to the system, each with its degree of freedom".

Authors’ reply: This is a valuable comment. However, our purpose was simply to make evidence of the reduction in Shannon dissipation due to the increase in E-particles, it is not a precise calculation of entropy in the whole system, as this is particularly cumbersome to do, yet.

At the end of section 2, where intracellular confined water is mentioned [refs 55-57], the works by Spitzer should be cited: J. Spitzer. From water and ions to crowded biomacromolecules: In vivo structuring of a prokaryotic cell. Microbiol. Mol. Biol. Rev. 75: 491–506 (2011), and J. Spitzer & B. Poolman. The role of biomacromolecular crowding, ionic strength, and physicochemical gradients in the complexities of life’s emergence. Microbiol. Mol. Biol. Rev. 73: 371–388 (2009).

Authors’ reply: References added accordingly

The first sentence in section 3 (lines 302-304) reads as a somewhat too simplistic view: "... the dynamics of a living object, called organism." Since the living organism is intricately connected to its environment, which has been discussed previously, one would expect here the mentioning of at least the interaction of this organism with its environment, if not replacing the living organism with a living ecosphere.

Authors’ reply: The sentence was revised, accordingly

Minor:

Abstract, line 14: theses (not thesis)

Authors’ reply: Done

Line 67: We have to thoroughly revise (rather than has to)

Authors’ reply: Done

Line 105: matter (rather than matters)

Authors’ reply: Done

Line 169: is much too high (rather than is too much high)

Authors reply: Done

Line 293: It is presumed that (rather than presumable)

Authors’ reply: Done

Reviewer 4 Report

This is a work that represents the point of view of the authors in the, difficult to define and discuss, origin of life. The work is so wide, and discussion of the different points is necessarily so interlinked that is difficult to differ as referee in some points without to arise multiple and complex questions in other points.

This is a type of work that is accepted or rejected in the basis of editorial decision, because being sound arguments there belong in very high percentage to the range of metaphysical proposals, therefore, the author responsibility on the published text is fully and low for reviewers.

I place too the question if MOLECULES is the right journal for this when mdpi have publications devoted to the life topic.

In any case the authors should take into the following commnents before publication:

  • The signs between Shannon and state function entropy (lines 216-220 compared to 239-240). Information follows the inverse path, such as the authors comment (lines 237-239).
  • For this referee the thermodynamic description of dissipative systems published by Glandsdorff-Prigogine scientific framework where all life models should be adjusted. In this respect, in Fig. 1 I think the flows of the called Boltzmann entropy and Shannon entropy should not follow the same direction. The Boltzmann entropy increase in the universe but must decrease in the life system as well Shannon entropy increases. The question is that the balance between entropy flow rates of the inside towards the outside must be achieved and the entropy production rates are not an extensive state function as the Boltzmann entropy. However, this is a so extensively overlooked question, that should not be hinder the publication. However, perhaps the authors could improve Fig. 1 and the corresponding text discussion.  

Author Response

Reviewer 4

This is a work that represents the point of view of the authors in the, difficult to define and discuss, origin of life. The work is so wide, and discussion of the different points is necessarily so interlinked that is difficult to differ as referee in some points without to arise multiple and complex questions in other points.

Authors’ reply: we meet this valuable comment and tried to better focus and explicate our theses.

This is a type of work that is accepted or rejected in the basis of editorial decision, because being sound arguments there belong in very high percentage to the range of metaphysical proposals, therefore, the author responsibility on the published text is fully and low for reviewers.

Authors’ reply: We are fully aware of this

I place too the question if MOLECULES is the right journal for this when mdpi have publications devoted to the life topic.

Authors’ reply: This should be an issue that Editors of Molecules can address. I am a Member of the Editorial Board of Molecules, and in my opinion the origin of life may perfectly fit the topics dealt in the journal. However, it is only an opinion of mine…

In any case the authors should take into the following commnents before publication:

  • The signs between Shannon and state function entropy (lines 216-220 compared to 239-240). Information follows the inverse path, such as the authors comment (lines 237-239).

Authors’ reply: I checked and equations are correct as they refer simply to Shannon entropy, as is. The minus sign refer to logarithmic property.

  • For this referee the thermodynamic description of dissipative systems published by Glandsdorff-Prigogine scientific framework where all life models should be adjusted. In this respect, in Fig. 1 I think the flows of the called Boltzmann entropy and Shannon entropy should not follow the same direction. The Boltzmann entropy increase in the universe but must decrease in the life system as well Shannon entropy increases. The question is that the balance between entropy flow rates of the inside towards the outside must be achieved and the entropy production rates are not an extensive state function as the Boltzmann entropy. However, this is a so extensively overlooked question, that should not be hinder the publication. However, perhaps the authors could improve Fig. 1 and the corresponding text discussion.  

Authors’ reply: We meet this valuable observation. However, we are simply reporting that when an event space, made of particles, each bearing a Shannon information, increases the amount of “equal” particles, the Shannon entropy decreases. There ae more particles having the probability to reach correctly a receiver (reiterated particle) and/or there is much less crowded different messages in the event space, so Shannon entropy lowers.  

Round 2

Reviewer 1 Report

This paper has several interesting ideas, and although speculative, I would like to see it published. However, even after the authors' latest revision, the paper is still rambling and not focused. It is hard to read and understand, and the main points are not emphasized appropriately. Additionally, the English is full of typos and spelling mistakes. So I still cannot recommend publication in its present form.

I might suggest that the authors show the paper to other colleagues or technical writers for help in writing and organizing the paper.

The paper discusses the mechanism of "Shannon dissipation". In a thermodynamic dissipative system, the increase in entropy outside the system allows for its decrease inside the system; this is governed by the Second Law of Thermodynamics. In the parallel, information driven Shannon Entropy, is there a similar law that governs its behavior? Maybe there is, but I did not succeed in understanding this from the paper.

I further suggest the authors carefully look again at the paper by Mayer that I previously mentioned, presently Ref. 7, since I think some of the basic ideas of this manuscript may be found in Mayer's paper. At first glance, they might seem different. But considering that Kolmogorov Complexity has been shown to be equivalent to Shannon Entropy (see, for example, K. Leung-Yan-Cheong and Cover, IEEE Trans. Information Theory, IT-24, 331, 1978), Mayer's description of the roles played by Order and Complexity in life seems particularly relevant.

Author Response

Point by point rebuttal to the Reviewers’ comments

Round 2

Reviewer 1

This paper has several interesting ideas, and although speculative, I would like to see it published. However, even after the authors' latest revision, the paper is still rambling and not focused. It is hard to read and understand, and the main points are not emphasized appropriately. Additionally, the English is full of typos and spelling mistakes. So I still cannot recommend publication in its present form.

Authors’ rebuttal: We are particularly thankful to this Reviewer’s comment, as it provided us with the opportunity to improve our writing style and reading of the manuscript. We therefore met these recommendations and tried to better clarify the meaning of our report, which was revised accordingly.

I might suggest that the authors show the paper to other colleagues or technical writers for help in writing and organizing the paper.

Authors’ rebuttal: We did and who participated in the revision has been acknowledged

The paper discusses the mechanism of "Shannon dissipation". In a thermodynamic dissipative system, the increase in entropy outside the system allows for its decrease inside the system; this is governed by the Second Law of Thermodynamics. In the parallel, information driven Shannon Entropy, is there a similar law that governs its behavior? Maybe there is, but I did not succeed in understanding this from the paper.

Authors’ rebuttal: We comprehend this comment and therefore we tried to better elucidate this issue in the manuscript, without altering so much the texture of our writing. A dissipative system works out of the thermodynamic equilibrium, though it obeys to the second principle. The Boltzmann space obeys to the classical rules held by thermodynamics, whereas entropic changes in the Shannon space (embedded in the Boltzmann space) are driven by conformational (structural) changes in the intracellular and confined water. Some reference cited in the text shows that information changes, as occurring in different structural changes in the microstate, drive energy changes, i.e. thermodynamic events. According to our opinion, this mechanism is insufficient in pushing forward an evolutive dynamics, as it could be easily dampened by energy fluctuations in the microstate. We suppose that, as thermodynamics in the Boltzmann space needs a sink of increasing entropy to go ahead, the same should be for the Shannon space. We retrieved this “positive entropy sink” in the accretion of disquisotropy during the replication mechanism, whose ignition started with dissipation. Obviously, when a component of a population of objects in the Boltzmann space is amplified (by replication), entropy in Boltzmann increases, whereas entropy in Shannon decreases. The formation of polymers, for example, reduces the entropy in the Boltzmann space, whereas, as a polymer is a combinatorial variability (and contain a variable massage), Shannon entropy increases. Therefore, during the dissipation holding life, often Boltzmann and Shannon appear to work in opposite ways regarding entropy, despite they both draw energy from thermodynamics, though in different modalities.

I further suggest the authors carefully look again at the paper by Mayer that I previously mentioned, presently Ref. 7, since I think some of the basic ideas of this manuscript may be found in Mayer's paper. At first glance, they might seem different. But considering that Kolmogorov Complexity has been shown to be equivalent to Shannon Entropy (see, for example, K. Leung-Yan-Cheong and Cover, IEEE Trans. Information Theory, IT-24, 331, 1978), Mayer's description of the roles played by Order and Complexity in life seems particularly relevant.

We read carefully the recommended paper from Mayer, we cited in the manuscript and which we appreciated, but Mayer did not address either Shannon entropy or dissipation in his paper. We are persuaded that the mechanisms by which complexity arose in the evolutive pathway of life was arranged by dissipative phenomena, not merely by informatics or spontaneous order such as in mineral crystals (Mayer’s paper page 5, chapter 6). Therefore, the very interesting parallelism with the Kolmogorov complexity, regarding Shannon entropy, is out of our purposes.  We are aware that in any programming language we can define the Kolmogorov complexity as the length of the smallest program that generates a long, complex and casual string of characters (symbols), compelling the system to read the string letter by letter. So the complexity is defined with respect to a certain description language. This is true if the complex system we outlined in our manuscript has a mechanistic teleonomy. For infinite strings, these issues are a little more interesting because if one doen't have a program that will generate the string, anyone basically doesn't have the string in any practical sense. In a mechanistic view, without a program that generates the digits of an infinite sequence noone can actually define the string. The parallelism with Shannon entropy in the K Leung-Yan-Cheong’s paper is intriguing but may get us out of our logical tenet, misleading our meaning.

Reviewer 2 Report

The authors have appropriately answered my questions and also pointed out my misunderstandings. Furthermore, the authors have revised extensively the previous manuscript as accepting my comments. So, I consider that the revised manuscript is acceptable.

Author Response

Reviewer 2

The authors have appropriately answered my questions and also pointed out my misunderstandings. Furthermore, the authors have revised extensively the previous manuscript as accepting my comments. So, I consider that the revised manuscript is acceptable.

Authors’ rebuttal: We are delighted of this comment. Thank you